# Potential Modifying Effect of the *APOEε4* Allele on Age of Onset and Clinical Manifestations in Patients with Early-Onset Alzheimer’s Disease with and without a Pathogenic Variant in *PSEN1* in a Sample of the Mexican Population

**DOI:** 10.3390/ijms242115687

**Published:** 2023-10-28

**Authors:** César A. Valdez-Gaxiola, Eric Jonathan Maciel-Cruz, Rubiceli Hernández-Peña, Sofía Dumois-Petersen, Frida Rosales-Leycegui, Martha Patricia Gallegos-Arreola, José Miguel Moreno-Ortiz, Luis E. Figuera

**Affiliations:** 1División de Genética, Centro de Investigación Biomédica de Occidente, IMSS, Guadalajara 44340, Jalisco, Mexico; cesar_korn19@hotmail.com (C.A.V.-G.); ejmacielcruz@hotmail.com (E.J.M.-C.); rubiceli.hernandez3493@alumnos.udg.mx (R.H.-P.); sofiadumois@hotmail.com (S.D.-P.); frida.rosales9343@alumnos.udg.mx (F.R.-L.); marthapatriciagallegos08@gmail.com (M.P.G.-A.); 2Doctorado en Genética Humana, Centro Universitario de Ciencias de la Salud, Universidad de Guadalajara, Guadalajara 44340, Jalisco, Mexico; jmmo10@hotmail.com; 3Maestría en Ciencias del Comportamiento, Instituto de Neurociencias, Centro Universitario de Ciencias Biológicas y Agropecuarias, Universidad de Guadalajara, Guadalajara 44340, Jalisco, Mexico; 4Instituto de Genética Humana “Dr. Enrique Corona Rivera”, Departamento de Biología Molecular y Genómica, Centro Universitario de Ciencias de la Salud, Universidad de Guadalajara, Guadalajara 44340, Jalisco, Mexico

**Keywords:** early-onset Alzheimer’s disease, apolipoprotein E, age of onset, clinical manifestations

## Abstract

In Alzheimer’s disease (AD), the age of onset (AoO) exhibits considerable variability, spanning from 40 to 90 years. Specifically, individuals diagnosed with AD and exhibiting symptoms prior to the age of 65 are typically classified as early onset (EOAD) cases. Notably, the apolipoprotein E (APOE) ε4 allele represents the most extensively studied genetic risk factor associated with AD. We clinically characterized and genotyped the *APOEε4* allele from 101 individuals with a diagnosis of EOAD, and 69 of them were affected carriers of the autosomal dominant fully penetrant *PSEN1* variant c.1292C>A (*rs63750083*, A431E) (PSEN1+ group), while there were 32 patients in which the genetic cause was unknown (PSEN1− group). We found a correlation between the AoO and the *APOEε4* allele; patients carrying at least one *APOEε4* allele showed delays, in AoO in patients in the PSEN1+ and PSEN1− groups, of 3.9 (*p* = 0.001) and 8.6 years (*p* = 0.012), respectively. The PSEN1+ group presented higher frequencies of gait disorders compared to PSEN1− group, and apraxia was more frequent with PSEN1+/APOE4+ than in the rest of the subgroup. This study shows what appears to be an inverse effect of *APOEε4* in EOAD patients, as it delays AoO and modifies clinical manifestations.

## 1. Introduction

Alzheimer’s disease (AD) is defined by progressive and irreversible changes that involve declines in cognitive functions, language behavior, and judgment, as well as memory loss [1]. AD’s neuropathological particularities include senile plaques, which represent the accumulation of beta-amyloid protein (Aβ) resulting from the cleavage of amyloid precursor protein (APP) [2]. Recent estimates indicate that around 50 million people worldwide suffer from dementia, and this number is expected to triple by 2050. AD is the most prevalent cause of dementia, and it is estimated that approximately 6.5 million people in the United States are currently living with AD [3]. In most cases, AD has no clear inheritance pattern, and the average age of onset (AoO) is 65 years [4].

An atypical form of AD known as early onset Alzheimer’s Disease (EOAD) affects individuals before the age of 65 years. This condition has a major genetic component and is typically regarded as a more severe variant of AD, characterized by an accelerated cognitive decline and a reduced life expectancy. Disease progression is usually rapid, with an average of 7.5 years until the patient dies [5,6,7,8]. A large number of EOAD cases have an autosomal dominant form of inherence; in addition, patients may have atypical manifestations such as cephalea, myoclonus, seizures, gait disorders, spasticity, and hyperreflexia, among others [9]. The exact incidence of EOAD is uncertain, but it is estimated to affect approximately 200,000 people in the United States [8].

The main genes affected are *PSEN1* (14q24.2), *PSEN2* (1q42.13), and *APP* (21q21.3) [4]. Variants in *PSEN1* disrupt the function of γ-secretase, in which PSEN1 functions as a catalytic subunit in the cleavage of the carboxyl-terminal fragment of APP, producing larger amounts of Aβ. The aggregation of Aβ in the brain parenchyma initiates a chain of events that ultimately leads to AD [10,11]. 

In 1999, a Mexican-American family with EOAD was identified in Southern California in the United States of America. The affected members were found to carry the c.1292C>A (*rs63750083*, A431E) pathogenic variant in the *PSEN1* gene. Further investigation revealed that an additional 24 families also had this variant, suggesting what could be a founder effect, likely originating in Mexico in the Los Altos de Jalisco region [12,13]. The families affected by this condition had an early AoO (with a mean age 41 years), and 40% presented gait disorders as an early manifestation [6,8,14]. To our knowledge, there are only four populations in Latin America with a high frequency of a pathogenic variant in *PSEN1*: a group of families from Antioquia, Colombia, which includes approximately 5000 carriers of the *rs63750231* (E280A) variant [15]; a large family from Cuba consisting of 281 members within six generations carrying the *rs63751144* (L174M) variant [16]; a group in Puerto Rico with more than 70 families affected by the variant *rs63750082* (G206A) [17]; and a group with the *rs63750083* (A431E) variant in the Mexican population, with at least 100 families affected.

The *APOE* gene is located at 19q13.32 and encodes for apolipoprotein E (APOE); the resulting protein has three isoforms which are transcribed by three polymorphic alleles, *APOEε2*, *APOEε3*, and *APOEε4*. APOE facilitates the transportation of cholesterol and other lipids to neurons in the brain while also playing an essential role in neuronal growth, neuronal plasticity, and membrane repair [18,19].

Regarding late onset Alzheimer’s Disease (LOAD), the *APOEε3* allele has been described as a “neutral allele”, whereas being heterozygous for the *APOEε4* allele increases the risk of developing AD by as much as threefold, and being homozygous for it increases the risk by up to tenfold. In contrast, individuals carrying the *APOEε2* allele have a 40% lower risk of developing AD, and recent studies indicate that individuals homozygous for *APOEε2* have an extremely low risk of developing AD in the absence of any additional risk factors [20,21].

Therefore, the purpose of this study was to identify the pathogenic variant in *PSEN1* c.1292C>A (*rs63750083*, A431E) and the *APOEε4* allele in all patients diagnosed with EOAD and determine its correlation with the AoO and clinical manifestations.

## 2. Results

From a total of 101 index cases evaluated for EOAD, 69 were positive for the pathogenic variant c.1292C>A (*rs63750083*, A431E) (PSEN1+ group) and 32 were negative (PSEN1− group). It should be noted that all patients were availed of our services after several years of disease progression. All cases were traced to an origin in the state of Jalisco at least three generations ago, with the exception of one case whose ancestors hailed from Mexico City.

### 2.1. Sociodemographic Information

Eighteen patients were observed in the PSEN1−/APOE4− group and fourteen in the PSEN1−/APOE4+ group. The mean ages of onset were 42.8 ± 10.5 and 51.4 ± 5.7 (*p* = 0.012) years, respectively. In the case of the PSEN1+/APOE4− (*n* = 59) and PSEN1+/APOE4+ (*n* = 10) groups, the ages of onset were 41.2 ± 3.5 and 45.1 ± 3.6 (*p* = 0.001) years, respectively. It has to be pointed out that all PSEN1+ group patients, despite the *APOEε4* result, presented an AoO <50 years (Table 1). In this study group, the *APOEε4* allele seems to delay the AoO. In the PSEN1− group, a mean AoO of 44.4 vs. 51.4 years was observed for the APOE4− and APOE4+ subgroups, respectively. In the PSEN1+ group, the means of AoO were 41.2 and 45.1 years, respectively (Figure 1). There were no statistically significant differences in disease duration among the subgroups. The mean number of years of education appears to decrease as genetic risk factors are added, but no statistically significant differences were found (Table 1).

### 2.2. Clinical Manifestations

The evaluated manifestations were classified according to behavior in the cognitive and motor spheres. Across all molecular subgroups, amnesia was the most frequent symptom. In our study group, irritability was more prevalent in the presence of APOE4+ or PSEN1+, and it was extremely prevalent (90%) when both genes were affected. Patients with PSEN1+ presented higher frequencies of gait disorders compared to those with PSEN1−, and apraxia was more frequent with PSEN1+/APOE4+ than in the rest of the subgroups. The frequencies of the clinical manifestations are presented in Table 2; manifestations less frequent than 10% in all subgroups were omitted. The most common neuroimaging finding was atrophy, which was observed in >60% of all subgroups, and 90% of the PSEN1+/APOE4+ patients presented with atrophy with or without vascular lesions (Table 2).

In our study population, the cognitive sphere was more affected in the PSEN1−/APOE4+ subgroup, and less affected in the PSEN1+/APOE4+ subgroup; in contrast, the behavior-sphere manifestations were more common in the PSEN1+/APOE4+ subgroup, and the less commonly affected subgroup was PSEN1−/APOE4+. Regarding the PSEN1−/APOE4− and PSEN1+/APOE4− subgroups, the proportion of affected spheres was similar. However, no statistically significant differences between subgroups and affected spheres were found (Figure 2).

## 3. Discussion

The aim of this study was to test the possible effect of the *APOEε4* allele in determining the AoO and clinical symptoms in EOAD patients with and without a pathogenic variant in *PSEN1*. Several studies have investigated the AoO and cognitive decline associated with the *APOEε4* status in LOAD [22,23,24,25]; however, the implication of *APOEε4* with or without pathogenic variants of *PSEN1* in EOAD has not been widely studied. To the best of our knowledge, this is the first study to compare two subpopulations of EOAD, one with a pathogenic variant in *PSEN1* and another in which the cause is not yet known.

The impact of APOE and AoO in EOAD has been relatively unstudied, and contradictory data have been found. In the study of Lendon et al., the findings suggest that *APOE* alleles do not seem to exert an impact on the age at which the disease starts [26]. Another study by De Luca et al. revealed that the *APOEε4* allele exerts an inverse impact on the AoO of the disease in LOAD and EOAD: a premature onset in LOAD, and a belated onset in EOAD [27]. In addition, Velez et al.’s results indicate that the *APOEε2* allele is associated with a significant delay in the AoO in individuals with the pathogenic variant *PSEN1* E280A [28].

Regarding the role of *APOE* and clinical manifestations in non-*APOEε4* carriers with EOAD, a faster decline in language and visuospatial functions has been reported compared to *APOEε4* carriers. Furthermore, non-*APOEε4* carriers with EOAD demonstrated a faster decline in Mini–Mental State Examination (MMSE) scores, indicating that patients without an *APOEε4* allele experienced faster deterioration in all cognitive domains except for memory [29,30].

Evidence shows that the *APOEε4* allele influences autosomal dominant EOAD, because in patients with pathogenic variants of *APP*, the *APOEε4* allele is associated with faster cognitive decline, whereas in carriers of pathogenic variants of *PSEN1*, an inverse effect is observed [31]. The above suggests a potentially inverse effect of *APOEε4* in EOAD compared with LOAD, as well as a different effect in EOAD depending on the genetic cause.

Most behavioral manifestations are reported to be more manageable and less detrimental than cognitive ones; the first type can generate stress and challenges for the caregiver and the family environment, without affecting the patient’s perception of their quality of life. On the other hand, amnesia can lead to frustration and anxiety, which negatively impacts the patient’s quality of life, sense of identity, and entire family environment [3].

In this study, we found a correlation between the AoO and the *APOEε4* allele: patients carrying at least one *APOEε4* allele showed a delay in the AoO in both PSEN1 + and PSEN1₋ groups by 3.9 and 8.6 years, respectively. Our findings are consistent with those of some previous studies reporting that the *APOE* genotype modifies the AoO in EOAD patients [27,28], and are also consistent with studies that show an effect on clinical manifestations [29,30]. Here, we show a higher frequency of behavior manifestations in patients with PSEN1+/APOE4+ and a lower frequency of cognitive manifestations than in the other subgroups. In addition, the PSEN1−/APOE4+ group showed the inverse result (Figure 2), suggesting that the variants in the *PSEN1* and *APOEε4* alleles have a different interaction compared with EOAD patients with whom the genetic cause is unknown, as it has been shown that patients with *APOEε4* show greater cognitive decline compared to patients without this allele in LOAD [32,33,34]. In addition, PSEN1+/APOE4+ patients present a greater homogeneity in the AoO, as shown in Figure 1. This study shows what appears to be an inverse effect of *APOEε4* in EOAD patients, as it delays the AoO and modifies clinical manifestations.

One possible explanation is that *APOE* could be age-related due to the shift in *APOEε4*’s effect from physiological to pathogenic in patients aged 50–60 years old [35,36]. Furthermore, APOE participates in numerous biological systems beyond just the cerebrovascular system, indicating the involvement of additional mechanisms. This is in accordance with a recent proposal that APOE exhibits contradictory functions in aging and neurodegeneration, based on mice investigations [37].

*APOEε4* has been suggested to modulate the activity of *PSEN1* in this regard; when there is a disruption in *PSEN1* caused by a specific genetic variant (such as the *PSEN1* variant), the presence of *APOEε4* could potentially interact with this alteration and have an additional impact on Aβ generation, thereby modifying the risk of developing AD [38].

However, more studies in different populations and larger samples are required to corroborate our results.

## 4. Materials and Methods

This study included 101 patients with EOAD diagnoses (43 females and 58 males) at the División de Genética at the Centro Medico Nacional de Occidente—Instituto Mexicano del Seguro Social (IMSS), Guadalajara, Jalisco, México, between 2012 and 2023. Genealogy and clinical data were obtained from nonaffected family members. The primary caregivers were interviewed to determine the AoO and the initial cognitive, psychological, motor, and other neurological symptoms that had manifested. Following a comprehensive and structured clinical history, molecular analysis was performed on the probands. Written informed consent was obtained from the primary caregiver/legal representative.

### 4.1. DNA Extraction and PCR-RFLPs Conditions

DNA extraction was carried out from peripheral blood samples using the salting-out method [39]. The PCR conditions included 200 mM dNTPs, 34 pmol of primers, 3.0 mM MgCl2, 1.5 U Taq polymerase (Invitrogen, Life Technologies, Carlsbad, CA, USA), and 10% DMSO in a final volume of 25 uL.

Primers had the following sequences: F4 5′-ACAGAATTCGCCCCGGCCTGGTACAC-3′ and F6 5′-TAAGCTTGGCAC GGC TGTCCAAGG A-3′ [40]. These primers were used to amplify a 174-bp fragment of the APOE gene. The PCR parameters were initiated by heating the mixture to 94 °C for 5 min, followed by 39 cycles of melting (94 °C, 15 s), annealing (55 °C, 28 s), and extension (72 °C, 45 s), and a final extension at 72 °C for 7 min.

To identify the alleles, the amplified product was subjected to a restriction enzyme analysis with *HhaI* (New England Biolabs, Beverly, MA, USA), following the manufacturer’s instructions. The samples were separated using 8% polyacrylamide gel electrophoresis (19:1) followed by silver staining [41], revealing 72, 48, 35, and 19 bp fragments for the ε4 allele; 91, 48, and 35 bp for the ε3 allele; and 91 and 83 bp for the ε2 allele [42]. The *APOE* gene was stratified to identify the *APOEε4* allele. Patients were classified as negative (APOE4−) or positive (APOE4+) based on the presence of at least one *APOEε4* allele (either homo- or heterozygous).

### 4.2. Sanger Sequencing

Sanger sequencing was utilized only for exon 12 of the *PSEN1* gene, wherein the variant c.1292C>A (*rs63750083*, A431E) is situated. ExoSAP-IT™ Express reagent (Applied Biosystems™, Foster City, CA, USA) was used to purify the PCR product. The sequencing reaction was conducted with the BigDye Terminator v3.1 Cycle Sequencing Kit, and the sequencing reaction was purified using BigDye™ XTerminator Purification Kit (Applied Biosystems™, Foster City, CA, USA). The conditions used conformed with those specified by the manufacturer.

### 4.3. Ethics

This study follows the Regulations of the General Health Law on Health Research [43] of the Declaration of Helsinki by the World Medical Association (with the latest modification being in October 2014), as well as the current national and international codes for good clinical research practices [44].

In accordance with the Regulations of the General Health Law on Health Research [43], this study is classified as Type I research or non-risk research, as it involves a study that will obtain DNA samples through venipuncture. The samples were stored with a folio number, and the obtained data was entered into a database. At all times, the data was kept strictly confidential.

### 4.4. Dementia Diagnosis

Dementia was clinically diagnosed through a consensus reached by the study neurologist and two physicians with a specialty in dementia, relying on neurological assessments, a thorough neurological examination, and a comprehensive review of relevant health and functioning information prior to the dementia evaluation, as well as observations of changes in cognitive function and behavior obtained through an informant interview. The diagnosis of dementia was made in accordance with the MMSE criteria. AD was diagnosed using the criteria provided by the National Institute of Neurological and Communicative Disorders and the Stroke/Alzheimer’s Disease and Related Disorders Association.

### 4.5. Statistical Analysis

Descriptive and comparative analysis for both sociodemographic and clinical data were performed on IBM SPSS Statistics v.29; graphics were created using RStudio v4.2.2. Fisher’s exact test or a chi-square test was used to compare frequencies; according with data normality, the *t*-Student test or Mann–Whitney U test was performed to compare the quantitative variables.

## 5. Conclusions

### 5.1. Limitations

Although interesting, our findings should be approached cautiously, and research extended to encompass a broader array of variables to furnish clinical substantiation for the comprehension of the possible involvement of *APOEε4* in autosomal dominant EOAD. The primary limitation of our study is the number of included patients. However, it is crucial to note that the variant c.1292C>A (*rs63750083*, A431E) is extremely rare in the global population.

Another constraint lies in the fact that we did not investigate alternative risk genes linked to AD. Indeed, multiple novel AD risk genes are proposed annually [45]; as a result, the potential impact of these alternate risk genes on AoO cannot be ruled out. These genes should be subjected to assessment in both EOAD and LOAD cases of Alzheimer’s disease with respect to AoO.

### 5.2. Future Research Directions

EOAD and its age of onset could encompass a range of aspects aimed at enhancing our understanding of AD, its genetic underpinnings, and potential interventions. Because studying diseases of an autosomal dominant nature (for example, autosomal dominant EOAD, as presented in this study) can provide insights into the pathophysiology of multifactorial diseases (such as AD), it is important to continue in the following potential directions.

A genetic variability inquiry would investigate the influence of other genetic factors beyond the primary causative mutation, and explore how other genetic variants, known as modifiers, may impact the AoO and clinical manifestations in EOAD. An epigenetic factor inquiry would study epigenetic modifications that could influence the AoO in EOAD, and explore DNA methylation, histone modifications, and non-coding RNA interactions that may contribute to the modification of clinical manifestations and AoO. An inquiry into longitudinal studies would conduct long-term follow-up studies on families with EOAD mutations to track the AoO across generations. This can help to identify patterns, trends, and potential factors influencing variability. A precision medicine inquiry would explore personalized approaches to treatment based on the AoO, clinical manifestations and genetic profile, as well as investigate potential targeted therapies that could delay or mitigate the effects of EOAD based on individual characteristics.

## Figures and Tables

**Figure 1 ijms-24-15687-f001:**
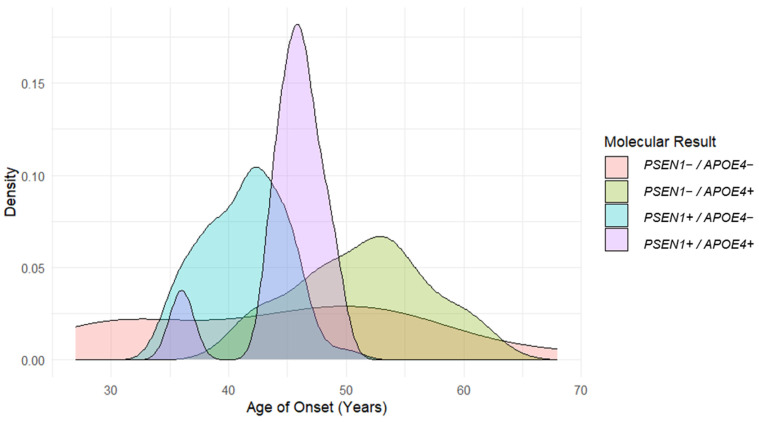
Density plot of age of onset according to molecular results of *PSEN1* and *APOEε4*. The first peak in the PSEN1+/APOE4+ subgroup (purple) corresponds to a single patient 38 years old. Patients PSEN1+ APOE4+ show a homogeneity in the AoO.

**Figure 2 ijms-24-15687-f002:**
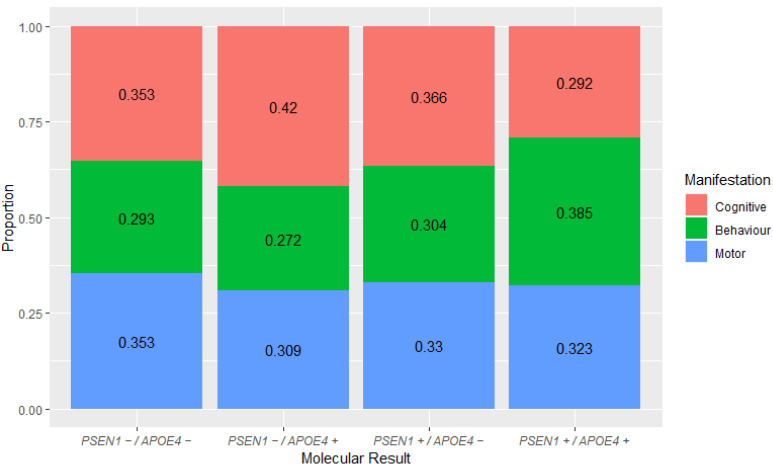
Proportion of clinical manifestations spheres vs. molecular results of PSEN1 and APOE4.

**Table 1 ijms-24-15687-t001:** Clinical information of dementia patients with PSEN1−/+ with or without *APOEε4* risk (*n* = 101).

Variable	PSEN1− *n* = 32			PSEN1+ *n* = 69		
	APOE4− *n* = 18	APOE4+ *n* = 14	*p*	APOE4− *n* = 59	APOE4+ *n* = 10	*p*
Sex (F/M)	9/9	7/7	1.000 ^a^	26/33	1/9	0.075 ^a^
Age of onset (years)	42.8 ± 10.5	51.4 ± 5.7	**0.012** ^b^	41.2 ± 3.5	45.1 ± 3.6	**0.001** ^c^
Disease duration (years)	6.4 ± 2.6	6.3 ± 3.2	0.855 ^c^	7.0 ± 2.9	7.9 ± 2.7	0.354 ^b^
Education (years)	11.5 ± 4.9	10.4 ± 3.7	0.667 ^c^	9.4 ± 3.8	8.2 ± 2.5	0.324 ^c^

F: female; M: male; APOE4−: without E4 alleles; APOE4+: with E4 alleles. a: Fisher’s exact test, b: t-Student, c: Mann–Whitney U.

**Table 2 ijms-24-15687-t002:** Frequencies of clinical manifestations according to molecular results of dementia patients with PSEN1−/+ with or without APOE4 allele (*n* = 101).

	PSEN1− *n* = 32		PSEN1+ *n* = 69	
	APOE4− *n* = 18	APOE4+ *n* = 14	APOE4− *n* = 59	APOE4+ *n* = 10
**Cognitive**				
Amnesia	16 (84%)	12 (86%)	51 (86%)	9 (90%)
Disorientation	8 (42%)	7 (50%)	23 (39%)	4 (40%)
Dyscalculia	6 (32%)	5 (36%)	23 (39%)	2 (20%)
Mutism	3 (16%)	2 (14%)	12 (20%)	3 (30%)
**Behavior**				
Irritability	9 (47%)	9 (64%)	38 (64%)	9 (90%)
Emotional lability	11 (58%)	6 (43%)	26 (44%)	5 (50%)
Insomnia	8 (42%)	5 (36%)	29 (49%)	5 (50%)
Depression	9 (47%)	2 (14%)	20 (34%)	3 (30%)
Anxiety	3 (16%)	3 (21%)	20 (34%)	3 (30%)
Aggressiveness	7 (37%)	5 (36%)	11 (19%)	3 (30%)
Hallucinations	4 (21%)	4 (29%)	14 (24%)	2 (20%)
**Motor**				
Dysarthria	8 (42%)	6 (43%)	39 (66%)	5 (50%)
Gait disorder	7 (37%)	2 (14%)	40 (68%)	9 (90%)
Apraxia	5 (26%)	2 (14%)	14 (24%)	6 (60%)
Stiffness	2 (11%)	0 (0%)	16 (27%)	5 (50%)
**Neuroimaging ***				
Atrophy	12 (67%)	11 (78%)	37 (62%)	6 (60%)
Vascular	6 (33%)	1 (7%)	8 (13%)	4 (40%)
Atrophy + Vascular	4 (22%)	1 (7%)	5 (8%)	1 (10%)
Normal	-	2 (14%)	1 (1%)	-
No data	4 (22%)	1 (7%)	18 (30%)	1 (10%)

APOE4−: without E4 alleles; APOE4+: with E4 alleles. * Includes CT or RMI.

## Data Availability

The data that support the findings of this study are available from the corresponding author upon reasonable request. Data available on request due to privacy/ethical restrictions.

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
