# Peer review of "Potential Modifying Effect of the APOEε4 Allele on Age of Onset and Clinical Manifestations in Patients with Early-Onset Alzheimer’s Disease with and without a Pathogenic Variant in PSEN1 in a Sample of the Mexican Population"

_ijms, 2023, doi:10.3390/ijms242115687_

Round 1
Reviewer 1 Report
" Potential Modifying effect of the APOEe4 allele on age of onset and clinical manifestations in patients with early- onset Alzheimer’s disease with and without a pathogenic variant in PSEN1 in a sample of Mexican population” by Valdez-Gaxiola et al. studied that functional variants in APOE and PSEN1 will affect the onset of Alzheimer’s disease. The idea of the study interests the reviewer and English is enough to publish it. However, the reviewer does not agree to publish the study at this point.
Point important: The Journal of International Journal of Molecular Genetics is a very important Journal at this point, and its impact factor is now 6.2 (2023). However, the sample size of this study is only101 even if the patients with early-onset Alzheimer’s disease were rare. In these days, this is too small to analyze in the genetic association study. That is because the effect of APOEe4+ allele was reversed.
Minor Point: Were the participants diagnosed with brain CT or MRI? It was kind of the authors to indicate it.
The reviewer thinks this study will give the Readers wrong information about the genes. Then, this article should be rejected.
Author Response
Dear Reviewer,
I hope this letter finds you well. I am writing to address the concern you raised regarding the limited sample size of 101 in our recent study. Your observation is indeed valid, and we appreciate your thoughtful consideration of our research.
The primary reason for the restricted sample size is the rarity of the subject population under investigation it is important to highlight that the variant studied is extremely rare due to its global frequency of 0.000008 reported by TOPMed (https://www.ncbi.nlm.nih.gov/snp/rs63750083#frequency_tab). As you may be aware, our study focused on individuals with early-onset Alzheimer's disease caused by a specific pathogenic variant. This subtype of the disease is exceedingly rare, making it challenging to identify and recruit a substantial number of participants. We concur that a larger sample size would have been ideal for certain types of analyses, particularly those related to associations.
However, it is important to emphasize that the primary goal of our study is descriptive and comparative in nature, rather than association-based, as the title suggests. Given the nature of our research, the pool of eligible participants was inherently limited. Our team dedicated extensive efforts and resources to locate individuals who met the specific criteria for inclusion in the study, given that it took us 10 years to reach this number of patients.
Nevertheless, it's important to note that our study is not the first report. We have taken into account the existence of other studies that have reported similar findings and have faced comparable challenges regarding sample size. The convergence of outcomes across studies with similar participant numbers underscores the consistency of our findings, even in the presence of a limited sample (e.g., doi: 10.1111/ene.15536; doi: 10.1111/ene.15536; doi: 10.1016/j.euroneuro.2015.03.014).
While we understand your concerns about potential confusion among readers, we are committed to ensuring that our results and conclusions are presented in a manner that aligns with the descriptive and comparative nature of our study.
Regarding the potential for result confusion, we understand the importance of presenting our findings in a clear and unambiguous manner. We are committed to addressing any ambiguity that might arise from our study. In the manuscript, now we included a dedicated section that discusses the limitations of our research, including the restricted sample size and its implications. We are also mindful of the responsibility to avoid misleading or confusing our readers. Our statistical analyses were conducted with full transparency, acknowledging the limitations inherent to the small cohort. Our aim is to contribute to the scientific discourse while maintaining transparency and precision in our reporting.
In light of these constraints, we believe that our study still holds value in contributing to the understanding of this rare form of early-onset Alzheimer's disease. Our findings, despite the limited sample size, provide valuable insights into the clinical presentation and genetic basis. In the case of the minor point raised regarding diagnostic neuroimaging, it is important to note that not all patients received this diagnosis, that's why further details are not included on this matter. However, among those who did, there is clear evidence of impairment.
Thank you for your time and consideration. We value your input and remain open to any suggestions or recommendations you might have to enhance the quality and impact of our study. Your expertise is invaluable to us, and we are grateful for the opportunity to contribute to the scientific community through your esteemed journal.
Reviewer 2 Report
The authors aimed to clinically characterize and genotype the APOEε4 allele from 101 individuals with a diagnosis of EOAD, 69 of them were affected carriers of the autosomal dominant fully penetrant PSEN1 variant c.1292C>A (rs63750083, A431E) (PSEN1+ group) and 32 patients which genetic cause is unknow (PSEN1- group). Interesting topic and good results part offering the perspective of a promising paper. Overall, this paper is written in a concise and orderly manner with sufficient introduction, detailed methods and solid data. The article is easy to read, well designated and presented, and can be of interest to reader and researchers. However, I have the following suggestions related to the improvements that should be added:
- Please specify the statistical analysis methods used.
- What would be the limitations of the present study? Possible future research directions?
- Please insert the appropriate references in the paragraph between the lines 219-221.
English is good, but there are spelling, punctuation and some grammar issues. This will apply to the whole manuscript.
Author Response
Dear Reviewer,
I hope this letter finds you in good health. I am writing in response to your recent feedback on our manuscript. We appreciate the time and consideration you have given to our work and would like to provide clarification on the concerns you raised regarding with the following points:
- Please specify the statistical analysis methods used.
We have added section 2.5 "Statistical analysis", where the methods used and the software employed are described.
- What would be the limitations of the present study? Possible future research directions?
We have also added section 5, "Limitations," and section 6, "Future Research Directions," in which the study's limitations are detailed, and what remains to be done in the research line is outlined.
- Please insert the appropriate references in the paragraph between the lines 219-221
We have included the appropriate citations for the mentioned articles.
- English is good, but there are spelling, punctuation and some grammar issues. This will apply to the whole manuscript.
We have meticulously reviewed the entire document and made the necessary revisions.
Thank you for your consideration, and we look forward to any further insights you may have to offer as we continue to refine our manuscript.
Round 2
Reviewer 1 Report
The sample size may be not enough to indicate the evidence. However, it will be informative for the Readers of the Journal. So, the manuscript is suitable for publishing the Journal.